# COVID-19, individual wellbeing and multi-dimensional poverty in the state of South Australia

Udoy Saikia⬤*, Melinda M. Dodd⬤, James Chalmers, Gouranga Dasvarma, Susanne Schech

College of Humanities, Arts and Social Sciences, Flinders University, Adelaide, South Australia

* udoy.saikia@flinders.edu.au

## Abstract

Research efforts in the initial months of the COVID-19 pandemic focused on the actual and potential impacts on societies, economies, sectors, and governments. Less attention was paid to the experiences of individuals and less still to the impact of COVID-19 on an individual's wellbeing. This research addresses this gap by utilising a holistic wellbeing framework to examine the impact of COVID-19 on the overall wellbeing of individuals in the Australian state of South Australia through an online survey. The research framework for the survey comprises six dimensions: psychological and emotional health, physical health, living standards, family and community vitality, governance, and ecological diversity and resilience. The results show that most respondents (71%) were able to maintain overall wellbeing during the pandemic. However, more than a half of the respondents could not maintain wellbeing in psychological and emotional health. Further examination of the drivers of inability to maintain overall wellbeing reveals that low-income individuals, younger respondents (aged 18–24) and women suffer disproportionate hardships. Defining poverty in terms of multidimensional deprivations in wellbeing enables a nuanced analysis of the unequal impacts of COVID-19 mitigation policies that can be used to improve policymaking.

## Introduction

Studies have shown that the COVID-19 pandemic, consistent with previous pandemics, is negatively and disproportionately impacting on particular groups of people, such as older adults, migrant workers, women, healthcare workers, and overseas students [1–3]. Further, the United Nations Department of Economic and Social Affairs [4] has noted early evidence which indicates that the health and economic impacts of the virus are borne excessively by the poor people; youth who are disproportionately vulnerable to employment loss; and those others who are employed but often work in the casual job market on contracts or in economic sectors severely affected by COVID-19. While some of these studies have focused on the psychological impact of the disease, including mental ill-health, stress, anxiety, and/or depression, others have considered economic, health, and/or educational factors of wellbeing.

**Data Availability Statement:** The participants did not consent to sharing the data publicly with other researchers/organisations, so the Flinders Research Ethics and Compliance has determined

that the data cannot be made publicly available, even in a deidentified form. Data access requests may be directed to Flinders Research Ethics and Compliance (contact via human. researchethics@flinders.edu.au).

**Funding:** The research was funded by Flinders University, Australia (https://flinders.edu.au/) through Flinders University Covid Research Grant 2020 with the approval from Flinders University Human Research Ethics Committee, Project ID: 2192, Project Title: Impact of COVID-19 on the wellbeing of individuals in South Australia. The funder had no role in study design, data collection and analysis, decision to publish, or preparation of the manuscript.

**Competing interests:** There is no competing interest that interferes with, or could reasonably be perceived as interfering with, the full and objective presentation, peer review, editorial decision-making, or publication of this article submitted to PLOS ONE.

However, these studies have tended to examine the impact of the pandemic on people with respect to each of these factors separately and for specific groups of people.

In contrast, the present research examines the impact of COVID-19 on the overall wellbeing of individuals. For the purpose of this study, the researchers have adopted an understanding of wellbeing derived from Amartya Sen's [5] ideas and have defined wellbeing as 'the freedoms and capability to make choices and act effectively with respect to, for example, health, education, nutrition, employment, security, participation, voice, consumption and the claiming of rights' [6 p1009]. Relatedly, wide-ranging international research informed by Sen's work has conceptualised poverty as multi-dimensional 'ill-being' that comprises material lack, physical ill-being, psychological ill-being, social exclusion, insecurity and powerlessness [7]. Building on Sen's ground-breaking conceptual work, and on the Gross National Happiness (GNH) index developed by the Oxford Poverty and Human Development Initiative [8], this research employs an innovative methodology based on the wellbeing framework developed by the authors (for detail please refer [9, 10]). The innovation of this approach is twofold, comprising a bespoke wellbeing framework and a set of measuring tools that capture objective and subjective indicators of wellbeing in the pandemic situation.

Tailored to the COVID-19 pandemic context, the framework accounts for wellbeing across multiple dimensions of individual experiences in the state of South Australia, one of six states in the Australian federation that have jurisdiction and responsibilities in key areas of wellbeing including health. Our framework includes both economic and non-economic impacts, with each dimension or domain composed of subjective and objective indicators. The research evaluated six domains of wellbeing: Psychological and Emotional Health; Physical Health; Living Standards; Family and Community Vitality; Governance; and Ecological Diversity and Resilience. While the first three domains are commonly linked to wellbeing, this framework also considers interpersonal connection and social engagement (Community Vitality), attitudes towards government oversight and actions (Governance), and activities involving nature and the environment (Ecological Diversity and Resilience) as important aspects of wellbeing. An attempt was made to also include Education as a distinct domain, however insufficient responses disallowed its inclusion in the wellbeing calculus.

The researchers hypothesised that the COVID-19 pandemic was having a profound impact on the wellbeing of individuals, and that the extent and type of the impact may be influenced by the socio-economic status of the area in which an individual resides, as well as key social factors such as age and gender. As such, the research aimed to address the following questions:

• What are the impacts of COVID-19 on the wellbeing of individuals in South Australia?

• Does the impact of the pandemic on wellbeing differ according to the socio-economic status, gender and age of individuals?

## Research design

### Materials and methods

The research received approval from the Flinders University Human Research Ethics Committee, Project ID: 2192, Project Title: Impact of COVID-19 on the wellbeing of individuals in South Australia. Data were collected and analysed anonymously. The survey was conducted online through a Flinders University dedicated website using Qualtrics survey software. The survey was only allowed for participants of age 18 and above. It was clearly mentioned on the cover page of the survey webpage. The aims and objectives of the survey were mentioned on

the cover page and consent of the participants were sought (by clicking "yes" or "no") before the participant could proceed to take part in the survey.

Due to the evolving COVID-19 situation at the time of data collection, a research design was developed that aligned with the authors' institution's COVID-19 research protocol involving human studies as well as Government advice and restrictions. This included avoiding face-to-face contact, undertaking online data collection, and ensuring a design that was flexible enough to accommodate the potential for additional restrictions over the course of the project. Approval for the research was granted by Human Research Ethics Committee of the research institution to which the authors are affiliated.

In previous studies, the authors of this paper adapted the wellbeing framework (see [9]) to study people's experiences of change in developing countries. These earlier studies were undertaken for preparing Human Development Reports in these countries on behalf of the United Nations Development Programme and were based on extensive household surveys.

For data collection in this study, an anonymous, voluntary, online survey was conducted using Qualtrics (https://www.qualtrics.com/au/core-xm/). The survey questions were developed based on the authors' wellbeing framework and adapted to South Australian conditions.

The Alkire-Foster methodology [11] is the basis of the current measurement. In this system, the step of identifying who falls into the category of "maintained wellbeing" or "not maintained wellbeing" relies on two cut-offs: one within each domain to determine the category (maintained or not maintained) with respect to each variable in that domain and a second cut-off (a nominal one) that is applied across domains. This nominal cut-off identifies individuals who are in the category (maintained or not maintained) sufficiency in wellbeing by counting the domains in which a person has achieved sufficiency in maintaining wellbeing.

To measure wellbeing at each domain, a two thirds rule is applied for calculations at each of the six domains. For example, there are 9 variables in the ecology domain. This means that the maximum score an individual could obtain in this domain is 9 (that is, if the participant scores 1 in each variable), and the minimum is 0 (that is, if the participant scores 0 in each variable), where the scores 1 and 0 denote achievement or non-achievement of sufficiency in a variable. By the two thirds rule, an individual (the participant) would need to score a minimum of 6 to be considered as having achieved sufficiency in maintaining wellbeing in this domain. The domain is then recoded as 0 for those who scored less than 6, and 1 for those who scored 6 or more. Thus, the number of respondents would be distributed between a score of 0 and a score of 1. In this way, each domain has a distribution of respondents with a score of either 0 or 1.

The level of overall wellbeing achievement (for all domains combined) of an individual is obtained by adding up the domain scores. This will be referred to as the composite Wellbeing Index. The two thirds rule is applied again here to determine whether the individual has maintained overall wellbeing or not. An individual is defined as "maintained overall wellbeing" if that individual scored 1 each in at least any four domains out of the total of six domains.

## Participants and distribution and data collection

Residents of South Australia aged 18 years or more were invited to complete an online survey in a six-week period from 3 August to 14 September 2020. During this time, the survey was promoted via social media as well as other promotion activities such as publicising on websites of amenable municipal Councils of the state, print and radio media interviews and announcements; placement on the researchers' University research webpages, and reposts via personal and professional networks. Participants were self-selected by opting to complete the survey comprising 56 questions on the six wellbeing domains.

The total survey sample size of 579 responses is statistically representative of South Australia's population aged 20+ with 99% confidence and within a 5% margin of error based on 2016 population data [12]. Of these, 442 responses, or 76%, were from the Greater Adelaide area, which corresponds with 2016 population data indicating that 77.2% of South Australia's population aged 20+ resides within Greater Adelaide. This distribution of responses allows for 95% confidence within a 5% margin of error that the Greater Adelaide sample is representative. The 81 responses from regional South Australia comprise 14% of the total responses, whereas the population of regional South Australia is 22.8% of the total State. As such, the regional South Australia sample is representative with 90% confidence within a 9% margin of error. The remaining 56 respondents, or 10%, did not indicate their area of residence.

Women comprise the vast majority of those who took the online survey at nearly 74.8% of the total sample. Men accounted for 17.6% of the sample, while 44 respondents, or 7.6%, of the sample did not disclose their gender (6.9%) or selected 'Other' (0.7%), which could be inadvertent or deliberate. As it is uncertain how the Australian general public responds to a gender option of 'Other' in these contexts [13], the authors determined it was not possible to draw conclusions for this group without further information.

In the following sections we discuss first the overall findings with a focus on the majority that maintained wellbeing, and then drill down into the factors that impact most on those who have not been able to maintain wellbeing.

## Wellbeing during COVID-19 in South Australia

The results (Table 1) show that 71% of respondents were able to maintain overall wellbeing as indicated by their wellbeing status in at least four of the six domains (two third rule as explained in Research Design section above). Wellbeing in the domain of Psychological and Emotional Health was the most difficult to maintain, with more than a half of the respondents (56%) reporting their inability to maintain wellbeing in this domain. This is not an unexpected finding, as many people are known to experience high levels of anxiety and depression during and immediately after a pandemic such as COVID 19, although anxiety is expected to decline over time as measures to contain the virus take effect [14, 15]. Additionally, while over 80% of the respondents maintained wellbeing in the domains of Living Standard, Community Vitality, and Governance over one third of the respondents could not maintain wellbeing in Physical Health and Ecological Diversity and Resilience. In this study, the Ecology domain reflects a

**Table 1. Percentage distribution of respondents by gender and who were able/not able to maintain wellbeing.** South Australia. 2020.

| Wellbeing domain | All respondents (%) | | Men (%) | | Women (%) | | Other (%) | |
|---|---|---|---|---|---|---|---|---|
| | Able to maintain | Not able to maintain | Able to maintain | Not able to maintain | Able to maintain | Not able to maintain | Able to maintain | Not able to maintain |
| *Overall Wellbeing* | *71.2* | *28.8* | *64.7* | *35.3* | *70.7* | *29.3* | *90.9* | *9.1* |
| *Psychological Health* | 44.0 | 56.0 | 49.0 | 51.0 | 38.3 | 61.7 | 90.9 | 9.1 |
| *Physical Health* | 65.0 | 35.0 | 61.8 | 38.2 | 63.0 | 37.0 | 90.9 | 9.1 |
| *Standard of Living* | 83.5 | 16.5 | 79.4 | 20.6 | 83.1 | 16.9 | 97.7 | 2.3 |
| *Family and community vitality* | 87.6 | 12.4 | 77.5 | 22.5 | 88.9 | 11.1 | 97.7 | 2.3 |
| *Governance* | 82.0 | 18.0 | 82.4 | 17.6 | 80.8 | 19.2 | 93.2 | 6.8 |
| *Ecological diversity and resilience.* | 64.8 | 35.2 | 54.9 | 45.1 | 64.2 | 35.8 | 93.2 | 6.8 |
| **Number of Respondents** | 579 | | 102 | | 433 | | 44 | |

Source: Computed by the authors in SPSS from results of *Impact of COVID-19 on the wellbeing of individuals in South Australia* 2020 survey.

complex systems perspective on wellbeing that highlights socio-economic circumstance, health and environment as complementary spheres. As such, the questions included in the Ecology wellbeing calculation focused on outdoor activity and engagement with nature in the very challenging circumstances presented by the COVID-19 pandemic (Table 1).

Wellbeing in Family and Community Vitality was maintained by the largest proportion of respondents, indicating high levels of support and engagement within families and larger social networks during the pandemic. Looking into the details of this dimension (not shown in Table 1), 62.4% of respondents were able to draw on the support of family and friends for help in dealing with difficult situations and/or to share worries and concerns. Approximately 77% kept in regular (at least once a week) contact with loved ones outside their household via telephone or other distance communication mechanism, while 50.5% kept in contact with their social network at least once a day. Additionally, over one half of all respondents checked on neighbours or people in their community who may have needed extra assistance.

However, there are considerable differentials between male and female respondents in the prevalence of overall and dimension specific wellbeing. For example, the prevalence of wellbeing is higher among women compared to men in terms of overall wellbeing, physical health, standard of living, family and community vitality, and ecological diversity and resilience, but women display higher levels of ill-being in psychological and emotional health, and governance (Table 1).

The respondents to the survey were asked to self-report the level of their positivity and hopefulness in life on a scale of 1 to 8, both before and at the time of survey during the COVID pandemic, where 1 denotes not positive and/or hopeful and 8 denotes very positive and/or hopeful. The results show that the percentage of respondents who felt highly positive and/or hopeful about life before the period of self-isolation/lockdown for COVID-19 fell by 28 percentage points during and after the COVID-19 self-isolation/lockdown period. Conversely, a much larger percentage of the respondents felt less positive/hopeful about life during the COVID period (Figs 1 and 2). Among all respondents, 62% reported that they are hopeful or satisfied with life in the COVID situation. This represents significant confirmation of the analysis where we found that 71% of individuals could maintain overall wellbeing.

While it has been possible to examine specific aspects of wellbeing within age groups during the COVID pandemic, determining the impact that age itself may have on wellbeing remains challenging. Despite evidence indicating that older persons are at higher risk of serious illness

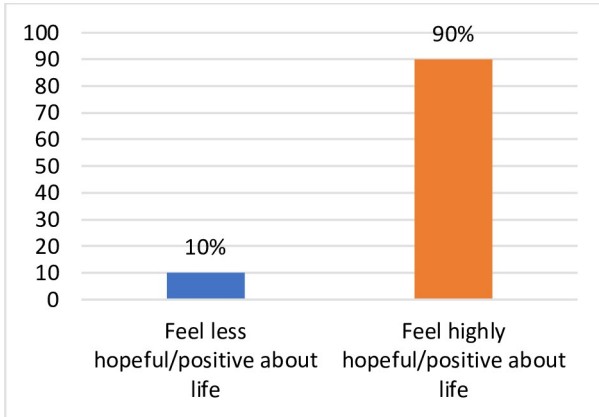

**Fig 1. Percent of respondents feeling positive/hopeful pre-COVID-19.** South Australia 2020. Source: Figures generated by the authors from results of *Impact of COVID-19 on the wellbeing of individuals in South Australia* 2020 survey.

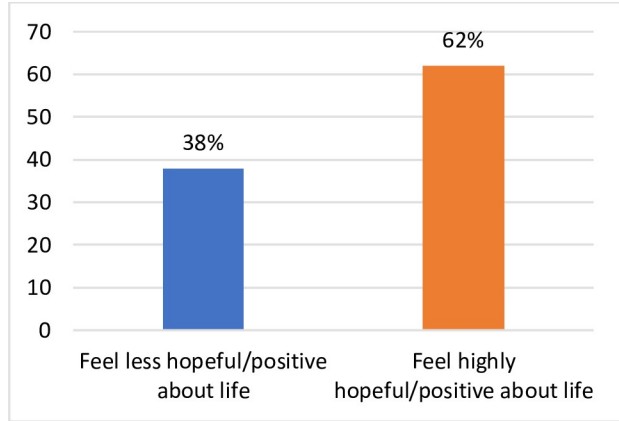

**Fig 2. Percent of respondents feeling positive/hopeful during COVID-19.** South Australia 2020. Source: Figures generated by the authors from results of *Impact of COVID-19 on the wellbeing of individuals in South Australia* 2020 survey.

and death from COVID-19, [16], a number of early studies have found younger adults are those most likely to be negative impacted financially and psychologically by the pandemic [17, 18]. A US study found that older adults aged 65+ are most likely of any age group to view COVID-19 as a threat to their health and least likely to consider it a threat to their personal finances while younger adults aged 18–29 view the threat to their personal finances as greater than that to their health [18]. Similar studies examining psychological distress during the COVID-19 pandemic found that younger adults were more than twice as likely than older adults to fall into the 'high distress' category, with one study noting that the same did not hold true in non-pandemic years [17, 18].

As shown in Figs 3 and 4, the results of this study indicate that both age and income were significant factors in maintaining overall wellbeing. Whereas 79% of respondents aged 65 and over maintained overall wellbeing, only 53% of respondents aged 18–24 were able to do so (Fig 3). Of those aged 18–24, 70% could not maintain psychological wellbeing and 53.3% could not maintain physical health wellbeing. However, a significant proportion same age group were able to maintain wellbeing in terms of Community Vitality (83.3%) and Governance (71.7%).

The results of this study also show a positive association between income and overall well-being, with the prevalence of maintaining overall wellbeing increasing with income (Fig 4).

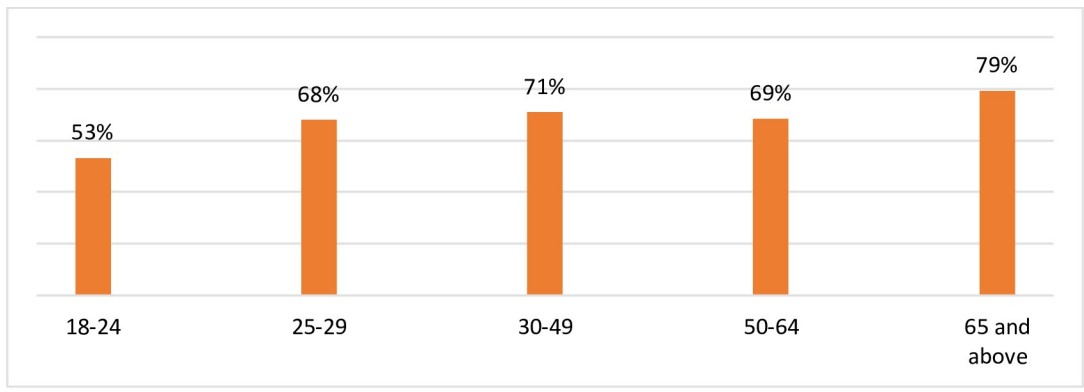

**Fig 3. Distribution of respondents maintaining wellbeing by age: South Australia 2020.** Source: Figures generated by the authors from results of *Impact of COVID-19 on the wellbeing of individuals in South Australia* 2020 survey.

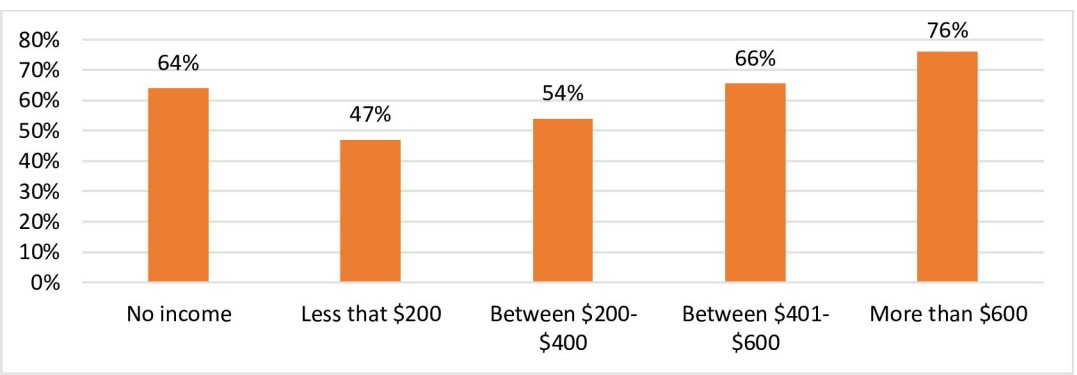

**Fig 4. Distribution of respondents maintaining wellbeing by income: South Australia 2020.** Source: Figures generated by the authors from results of *Impact of COVID-19 on the wellbeing of individuals in South Australia* 2020 survey.

This is consistent with the finding of other early studies that have found that those with higher levels of income have been less impacted by the COVID-19 pandemic, including indications that the mitigation measures put in place in Australia and elsewhere are causing low-income individuals to suffer disproportionate hardships [19, 20].

Similar studies in developed nations have found that lower-income adults are less pre-pared financially to withstand challenges such as job loss, sickness, or economic downturn that may occur as a result of COVID-19 than those with higher incomes [20, 21]. Related findings in the United States indicate that 44% of lower-income adults have had to utilise retirement or savings funds to pay bills since COVID-19 began, as compared with only 15% of upper-income adults [22]. These findings, as well as those of this study, become particu-larly relevant when viewed within the wider framework of the link between income and wellbeing. Previous studies into this link have posited that it is external factors influencing income, such as changes in income, rather than the level of income itself that are what impact wellbeing and that 'financial security has nearly three times the impact of income alone' on overall wellbeing [23, 24 p6].

This may provide a partial explanation for the unexpected observation of the "no income" group showing relatively higher percentage maintaining wellbeing (Fig 4). A large proportion (83%) of participants in this group belong to the older cohort (age 50 and above), the majority of whom may depend mainly on old age pensions that provide financial security, but are likely to have reported their income as "no income".

When asked about their 'attitude toward Government policies on managing the COVID-19 pandemic (i.e. social distancing, mask wearing, self-isolation, etc)', 92% of respondents rated their attitude as accepting. This is a higher positive response than that reported in an April 2020 survey of Australians, which found that while the majority of people were following dis-tancing requirements, 19.7% of those surveyed agreed or strongly agreed that 'There has been too much unnecessary worry about the COVID-19 outbreak' [25 p6]. The same study also found improvements in confidence in Government, with 66.7% of respondents reporting con-fidence in State/Territory governments specifically. This is consistent with the result of this research, in which 64.6% of respondents rated 'the overall performance of the South Australian Government in dealing with the pandemic situation' as high or very high.

The survey also collected information on the respondents' suburb and Council area. Although the number of responses for any one suburb or Council was not large enough to pro-vide a representative sample size, it was possible to compare the experience of regional South Australians with those living in the Greater Adelaide area. This analysis shows that residents of

the Greater Adelaide area were more likely than those in regional South Australia to maintain overall wellbeing (70.2% vs 65.5%), but only marginally (Table 2).

It is hypothesised that a person's individual characteristics such as individual income (prior to pandemic), attitude towards life (prior to pandemic), age and gender would influence the likelihood that a participant maintained overall wellbeing during the pandemic. The dependent variable, namely the ability to maintain overall wellbeing, is conceptualised as a binary variable: 'able to maintain wellbeing' or 'not able to maintain wellbeing'. Therefore, a logistic regression analysis was performed to ascertain the effects of these characteristics on the respondents' overall wellbeing.

Table 3 shows the results of the binomial logistics regression analysis in which we estimate the probability of a South Australian person maintaining wellbeing. The aim is to calculate an odds ratio concerning the ability to maintain wellbeing in the presence of the following selected variables identified above: weekly income, hopefulness about life, age, and gender. The idea is to find a relationship between these explanatory aspects and the probability of maintaining wellbeing. For weekly income, the data was regrouped into two: i) $600 or less and ii) more than $600. The reason behind this regrouping is that the data show almost equal distribution of respondents between these two groups. Age was forumlated as presented in Fig 3. Hopefulness about life and gender were calculated as binary variables (hopeful/not hopeful and male/female, respectively).

In sum, logistic regression helps estimate whether the ability to maintain wellbeing is influenced by any of the selected variables mentoned above. In the binomial logistic regression the dependent variable is the ability to maintain wellbeing which is formulated as a binary variable. It takes the value 1 if the individual successfully maintains wellbeing, and 0 if the individual is not able to maintain wellbeing. The independent, or explanatory variables are weeekly income, hopefulness about life, age, and gender. The regression model explains 88.0% (Nagelkerke $R^2$) of the variance in maintaining wellbeing and correctly classifies 70.8% of cases. While age (p = .020), individual weekly income prior to pandemic (p = .003) and hopefulness about life prior to the pandemic (p = .001) add significantly to the model/prediction, gender (p = .166) does not. The results of the logistics regression analysis (Table 3) show that higher age, higher individual income and a more hopeful attitude to life are associated with an increased likelihood of maintaining wellbeing. While gender does not show any strong correlation with the ability to maintain overall wellbeing, however a larger percetage of women in the survey reported an ability to maintain wellbeing compared to men.

**Table 2. Wellbeing in Regional and Greater Adelaide areas at time of survey. South Australia 2020.**

|  | No of responses | response % |
|---|---|---|
| **Regional South Australia** |  |  |
| Could not maintain overall wellbeing | 28 | 34.5% |
| Maintained overall wellbeing | 53 | 65.5% |
| Total | 81 | 100.0% |
| **Greater Adelaide** |  |  |
| Could not maintain overall wellbeing | 132 | 29.8% |
| Maintained overall wellbeing | 310 | 70.2% |
| Total | 442 | 100.0% |

Source: Figures generated by the authors from results of *Impact of COVID-19 on the wellbeing of individuals in South Australia* 2020 survey.

**Table 3. Socio-economic determinants of wellbeing–results of binominal logistic regression.** South Australia 2020.

| Independent variables | B | S.E. | Wald | df | Sig. (p-value) | Exp(B) | 95% Confidence Interval (CI) for EXP (B) | |
|---|---|---|---|---|---|---|---|---|
| | | | | | | | Lower | Upper |
| Weekly income (Reference category: Less than $600) | 0.597 | 0.198 | 9.059 | 1 | 0.003 | 1.817 | 1.232 | 2.681 |
| Hopefulness (Reference category: 1, which is lowest in the scale of hopefulness in life before COVID) | 0.223 | 0.065 | 11.726 | 1 | 0.001 | 1.250 | 1.100 | 1.420 |
| Age (Reference category: lowest age group which is 18–24) | 0.189 | 0.081 | 5.373 | 1 | 0.020 | 1.208 | 1.030 | 1.417 |
| Gender (Reference category: Male) | | | 3.588 | 2 | 0.166 | | | |
| Constant | -2.716 | 1.252 | 4.703 | 1 | 0.030 | 0.066 | | |

Source: Figures generated by the authors from results of *Impact of COVID-19 on the wellbeing of individuals in South Australia* 2020 survey.

## Key factors in the inability to maintain wellbeing

To reiterate a fundamental finding of the main survey, the analysis showed that people who *could not maintain overall wellbeing* accounted for almost one-third of the total number respondents (167 of the total 579). This section focuses on these respondents to investigate which domains provided the greatest difficulty in maintaining wellbeing, and what factors affected their ability to adapt to the pandemic situation. Table 4 identifies the domains as the drivers of a lack of wellbeing in descending order of people's expressed dissatisfaction during the COVID-19 mitigation measures. It may be recalled that a person is deemed to have overall wellbeing if she/he could maintain wellbeing in at least four out of six domains. Thus, among those who could maintain overall wellbeing (a total of 412) there may be some who could not maintain wellbeing in one, or two or three domains. The third column of Table 4 shows these percentages. Based on identification of the drivers in Table 4, our analysis shows that the crucial question of poverty equates with multi-dimensional ill-being. This is taken up in the next several paragraphs, where poverty is understood not only in terms of income but more broadly in terms of what Frances Stewart [26] calls horizontal inequalities/ deprivation/ vulnerability.

Table 4 shows a nearly universal inability to maintain wellbeing with Psychological and Emotional Health among respondents during the period of COVID-19 mitigation measures.

**Table 4. Drivers of inability to maintain wellbeing among South Australians 2020.**

| Domain | Percentage (%) reporting inability to maintain wellbeing | |
|---|---|---|
| | Who could not maintain *overall wellbeing* | Who maintained *overall wellbeing* |
| Psychological and Emotional Health | 95 | 40 |
| Physical Health | 83 | 16 |
| Ecology wellbeing | 67 | 23 |
| Governance | 46 | 7 |
| Living standard | 45 | 5 |
| Community Vitality | 28 | 6 |
| Total number of responses (n) | 167 | 412 |

Source: Computed by the authors in SPSS from results of *Impact of COVID-19 on the wellbeing of individuals in South Australia* 2020 survey.

*Note*: percentages total more than 100 because they are based on multiple response.

World Health Organisation and other public health agencies have long understood that mental health is a risk factor for chronic physical conditions: those with severe mental health conditions are at high risk of experiencing protracted physical conditions; and people with chronic physical conditions are at risk of damaging their mental health [27–29]. Table 4 reconfirms this level of risk, showing just a 12 percentage-point difference in between the Psychological/ Emotional Health and Physical Health domains. The third high risk domain is Ecological wellbeing. More than two-thirds (67%) of respondents who were unable to maintain overall wellbeing reported inability to maintain wellbeing in this domain. Note that the inclusion of Ecology in this study is characterised by a complex systems perspective on wellbeing that highlights socio-economic, health and environment as complementary spheres. To this end, the questions in the Ecology domain probed outdoor activity and keeping fit in the very challenging circumstances of frustration, boredom, financial loss, joblessness, infection fears, inadequate supplies, inadequate nutrition, and inadequate information. In order to get a comparative picture, we have added, in Table 4 the drivers of inability to maintain wellbeing for the group of individuals "who were able to maintain overall wellbeing". While the general pattern of the inability to maintain wellbeing appears similar for both the groups, the considerable difference between the groups with respect to the physical health domain is indeed noteworthy.

For a more in-depth picture of deprivation of the people "who could not maintain wellbeing" (167 individuals), we examined the "intensity" of illbeing which indicates the average number of domains of dissatisfaction of these individuals. The intensity of ill-being is measured by the Alkire-Foster method [30], except that we did not have to consider weights for the different domains as we assigned equal weights to each domain. The breakdown of the 167 individuals by the number of domains of dissatisfaction shows that there are 4 individuals who were dissatisfied in all 6 domains, 16 dissatisfied in 5 domains, 61 dissatisfied in 4 domains and 86 dissatisfied in 3 domains. This gives the average number of domains of dissatisfaction as 3.6, which is the "intensity" of illbeing among the 167 individuals. In other words, the individuals in this group failed to maintain wellbeing in around four domains out of six, indicating that the intensity of illbeing is very close to the cut-off point of wellbeing.

## Income poverty and living standard

Of those who could not maintain wellbeing (193 respondents) 45% reported insufficiency in Living Standards during self-isolation. The depth of income poverty is important in relation to living standard, with just under two-thirds (63%) of these 193 respondents reporting weekly individual earnings of $600 or less. This amounts to less than *half* of the Australian average weekly total earnings for employees of $1,304 in May 2020 [31], and is well below the minimum wage of $753.80 in Australia [32].

Specific risks reported by income-poor people show that almost 90% of the respondents who were unable to overall wellbeing experienced reductions in, or stagnation of, their income and savings. More than two-thirds (70% and 71% respectively) reported a negative impact on their living standard and feeling less secure due to COVID-19. Nearly two-thirds of these income poor respondents were not able to soften the financial impact of the pandemic because they were not eligible for financial support from the Government such as JobSeeker or Job-Keeper payments (detail on these payment schemes can be found at Service Australia, Australian Government https://www.servicesaustralia.gov.au/). A small proportion (16%) were unable to pay rent.

Our data show that while inequality is partly captured by income poverty, its consequences extend into the domains of health and ecology. For example, poor health not only negatively

affects people's financial capacity if they have to withdraw from the labour force, but also generates financial poverty by increasing the amount of household expenditure on healthcare related items.

## Psychological and emotional health

Nine out of ten respondents who could not maintain overall wellbeing in the combined domains reported that they had suffered a 'negative impact' when asked the cross-checking question: *Overall, what kind of impact do you feel that the COVID-19 pandemic has on your mental health*?

This very large number of responses prompted further examination of the symptomatic/triggering factors of psychological and emotional ill-health, which are shown in Table 5 in descending order of the percentage of people answering "Yes" to the core questions canvassed in the Psychological and Emotional Health domain.

Inadequate physical activity is a salient issue. Seven out of ten of those unable to maintain overall wellbeing were not specifically engaging in physical activities. There are multiple health intersections that Table 5 identifies, such as sleep difficulties and concentration deficits. The finding on inadequate physical activity is reconfirmed in identical information for the Ecology domain discussed later in this paper.

Interpersonal conflict leading to violence is a particularly devastating impact of self-isolation. The public health information has been very clear on the risks of conflict or violence when people remain confined together for long periods [33, 34]. Our study shows that South Australia is no exception to this, with more than one-third of respondents (38%) in this group reporting an increase in interpersonal conflict in their household which is almost three times higher than the group who could maintain overall wellbeing. This has serious consequences for NGOs and public social workers who might be called in to attend to such conflicts or violence, especially since it heightens their own already high risk of infection.

## Physical health

Continuing with analysis of the domains that drive inability to maintain wellbeing, the Physical Health domain encompasses the second highest risk, with eight out of ten people with overall ill-being reporting their inability to maintain physical health wellbeing (Table 4). This finding was confirmed by the cross-checking question to which 83% of this group of respondents reported a negative impact of the COVID-19 pandemic on their physical health.

**Table 5. Mental health responses of those who could not maintain overall wellbeing, South Australia 2020.**

| Survey question on mental health condition | Percentage (%) reporting each condition |
|---|---|
| Depressed or anxious in the period of restrictions related to COVID-19? | 93 |
| Problems with concentration? | 86 |
| Problems with sleeping? | 78 |
| Not specifically doing physical exercise | 71 |
| An increase in interpersonal conflict in your household? | 38 |
| Total number of responses (n) | 167 |

Source: Computed by the authors in SPSS from results of *Impact of COVID-19 on the wellbeing of individuals in South Australia* 2020 survey.

*Note*: percentages total more than 100 because they are based on multiple responses.

A jarring finding is that six out of ten of respondents experienced an interruption to ongoing health related treatment due to government restrictions on movements and gatherings and despite the availability of tele-health options. This is an area that requires further study regarding the urgency or nature of these disorders, since it has been widely reported that a large proportion of elective surgery and procedures in Australia were deferred to free-up hospital spaces for anticipated COVID-19 patients., and a substantial drop occurred in hospital activity, across both public and private sectors [35].

Another risk factors (allowing for statistical error) that ranked alongside *interruption to ongoing health related treatment* relates to nutrition, and smaller, but still substantial proportions of respondents reported cutting down on food and health related expenditure (35%), but increasing alcohol or tobacco consumption (38%). This finding needs further study regarding its correlation with income-poverty, since it has become well-understood during COVID-19 that affordable nutrition is a crucial factor in enhancing the immune system, to protect the host from pathogenic organisms that include viruses [36].

## Ecological diversity and resilience

The ranking of the Ecology domain as a trigger of inability to maintain wellbeing is the third highest, after Mental Health and Physical Health. More than two-thirds (67%) of those who were unable to maintain overall wellbeing were not able to maintain wellbeing in the Ecology domain (Table 4).

As stated earlier, the Ecology domain is one of the top three drivers of the inability to maintain overall wellbeing. Delving into the underlying conditions, our findings show that lack of physical exercise is a specific high-risk factor, as is not utilising public parks or outdoor areas more than usual, eg. to compensate for the closure of sport facilities and restrictions in social activity. Notably, almost nine out of ten respondents in this grouping said they *did* have access to a nearby personal garden space. Notwithstanding this, they evidently prioritised other activities–an issue that needs to be further studied, for instance, to substantiate news reporting of the prioritisation of activities such as online gaming, social media use and/ or movie streaming [37–39].

## Gender disparities

We attempted to assess the gender gap among the survey population who reported they were unable to maintain wellbeing. Technically, this analysis is tentative rather than conclusive, because men comprise just 22% of the total 167 respondents who reported an inability to maintain wellbeing. Of the 579 respondents in total 102 were men, 433 women and 44 did not report their gender. Taking this in the context of gender distribution of the overall sample of 535 respondents who did report their gender. that consists of 19% men and 81% women, the higher percentage of men (22%) among the group of respondents reporting overall illbeing indicates that women are slightly better equipped to cope with illbeing in the COVID-19 environment.

A point that needs to be reiterated, as presented in Table 6, is that the Psychological and Emotional Health domain is of most serious risk to both genders. What can be added to by analysis of gender disparity is that women were disproportionately vulnerable. Almost *all* women (98%) in this cohort reported inability to maintain wellbeing in the domain of Psychological and Emotional Health, with a 12-gap, or 12.2% difference, between them and men.

Adding to this picture of psychological/emotional vulnerability are the earlier-noted findings on hopefulness, which respondents reported for the periods both before and during the COVID-19 lock-down. Hopefulness/positivity was high in the period before lock-down for

**Table 6. Gender disparities in psychological & emotional health among South Australians unable to maintain wellbeing.**

| Domain or Indicator | Total % of those unable to maintain wellbeing | Female respondents % of those unable to maintain wellbeing | Male respondents % of those unable to maintain wellbeing | Point differential between female & male respondents* | % difference (female to male respondents)* |
|---|---|---|---|---|---|
| Psychological/ Emotional health | 95 | 98 | 86 | 12 | 12.2 |
| Life expectations hopeful *before* COVID-19 | 14 | 14 | 11 | 3 | 21.4 |
| Life expectations hopeful due to COVID-19 measures | 71 | 72 | 61 | 11 | 15.3 |
| Sample size (n) | 163 | 127 | 36 | | |

*positive indicates greater vulnerability of women; negative (-) indicates greater vulnerability of men.

Source: Computed by the authors in SPSS from results of *Impact of COVID-19 on the wellbeing of individuals in South Australia* 2020 survey.

*Note*: percentages total more than 100 because they are based on multiple response.

women, with just 14% reporting low positivity, and only a 3-point gap separated them from men, of whom 11% reported low positivity: i.e.: both were positive. However, when asked if COVID-19 measures had impacted on their positivity, 72% of women reported a loss of hope, as compared with 61% of men. This represented a 11 percentage point (15%) gap between women and men in terms of loss of hopefulness.

However, our findings also record significant signs of male respondent vulnerability in the domain of Ecology. While almost two-thirds of female respondents reported inability to cope in the domain of Ecology, this was an 11-point (17.2%) stronger result compared to male respondents. This finding suggests that female respondents were more adept at finding ways to address the very challenging environments of lock-down associated with ecological wellbeing. The domain of Community Resilience shows further signs of more successful adaptation by women. While almost half of the male grouping were unable to maintain community belongingness, women reported a robust 18-point differential, equating to a 75% difference. This points to a much stronger capacity of women to draw on the support of others to help get them through the period of lock-down.

Notwithstanding the adaptive skills shown by women in this study, our findings show that almost half of the women distrusted the actions of government during the lock-down which is 8-points (17%) higher than men. The data also show an increase in household conflict, with 38% of those who failed to maintain wellbeing reporting an increase in conflict. Proportionately more women (41.7%) experienced an increase in interpersonal conflict in the household due to COVID-19 as compared to 30.7% of men. This was in response an explicit survey question: *did you experience an increase in interpersonal conflict in the household due to COVID-19*?

These findings become particularly salient when considered alongside other studies that have shown that violence against women, particularly domestic violence, has intensified since the onset of the COVID-19 pandemic [40, 41]. Further, it is well-understood that violence is an acutely damaging impact of self-isolation [33]. The broader implications of gender disparity and the intersection of these issues with regard to the impact of COVID-19 warrant further research.

## Conclusions and policy implications

The results of this study show that a majority of respondents were able to maintain overall wellbeing during the COVID-19 pandemic as experienced through mid-September 2020. Of

the six wellbeing domains that were measured, Family and Community Vitality appears to contribute the most to this achievement, followed by Standard of Living and Governance. Of particular note is that a significant proportion of respondents had an accepting attitude toward Government policies on managing the pandemic, and a majority rating the South Australian Government's overall performance in dealing with the pandemic as high or very high. While these results are broadly consistent with the findings of other Australian studies to date, as the pandemic continues further research and comparisons may be needed to draw longer term conclusions as to the perceived overall performance of Government.

While factors in the Standard of Living domain appear to have contributed to the achievement of overall wellbeing, the key message for policymaking concerns inequalities. Clearly, not everybody endured the same difficulties in the same way. The study confirms what has been well-understood in the research community, which is that the poorest members of society bear the brunt of the hardships in the time of COVID-19. Their experiences demand targeted decisions in order to prepare for remedial solutions in the time of the pandemic and to reduce suffering from future mitigation measures.

The more crucial message that this paper delivers relates to the findings showing a spike in household conflict and the areas in which women's wellbeing in particular, is at increased risk, i.e.: psychological/emotional health, physical health, and distrust in the actions of government. This increased risk is considerably higher among those who could not maintain overall wellbeing during the pandemic. As discussed previously, while this study asked specifically about interpersonal conflict, not violence per se, other studies have shown an increase in violence against women as a crucial impact of lockdown and that violence is an acutely damaging impact of self-isolation [33, 40].

Gender-sensitive measures that are needed in response to future pandemics and the likelihood of lock-down include boosting the provision of helplines and shelters. Additional cash transfers through JobKeeper and JobSeeker schemes need to target women with paid family and sick leave. Childcare services need to be enhanced. And ultimately, judicial responses are required to counter the escalation in violence against women and girls that occur during times of pandemic lock-down.

This research has demonstrated a valid method of examining the impact of the COVID-19 pandemic, and pandemics more broadly, on an individual's wellbeing in various dimensions of life. This has provided a scalable approach to studying pandemics and wellbeing in a variety of geographical contexts. More broadly, the findings of this research, and similar future studies utilising the study's methodology, can be used to support evidence-based policies for maintaining higher levels of wellbeing, both during a pandemic and in post-pandemic situations.

## Supporting information

**S1 File. Qualtrics survey software questionnaire.**
(PDF)

## Author Contributions

**Conceptualization:** Udoy Saikia, James Chalmers, Gouranga Dasvarma, Susanne Schech.

**Data curation:** Udoy Saikia, Melinda M. Dodd.

**Formal analysis:** Udoy Saikia, Melinda M. Dodd, James Chalmers, Gouranga Dasvarma.

**Funding acquisition:** Udoy Saikia.

**Investigation:** Udoy Saikia, Melinda M. Dodd, James Chalmers.

**Methodology:** Udoy Saikia, James Chalmers, Gouranga Dasvarma, Susanne Schech.

**Project administration:** Udoy Saikia, Melinda M. Dodd.

**Resources:** Udoy Saikia, Melinda M. Dodd.

**Software:** Udoy Saikia, Gouranga Dasvarma.

**Supervision:** Udoy Saikia.

**Validation:** Udoy Saikia, Melinda M. Dodd, James Chalmers.

**Writing – original draft:** Udoy Saikia, James Chalmers, Gouranga Dasvarma.

**Writing – review & editing:** Udoy Saikia, Melinda M. Dodd, James Chalmers, Gouranga Dasvarma, Susanne Schech.

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
