## [Decision Letter · Decision Letter 0]

27 Apr 2021

PONE-D-21-01771

COVID-19, individual wellbeing and multi-dimensional poverty in the state of South Australia

PLOS ONE

Dear Dr. Saikia,

Thank you for submitting your manuscript to PLOS ONE. After careful consideration, we feel that it has merit but does not fully meet PLOS ONE’s publication criteria as it currently stands. Therefore, we invite you to submit a revised version of the manuscript that addresses the points raised during the review process.

We look forward to receiving your revised manuscript.

Kind regards,

Rashidul Alam Mahumud, MPH, MSc, PhD

Academic Editor

PLOS ONE

Journal Requirements:

3. Please include additional information regarding the survey or questionnaire used in the study and ensure that you have provided sufficient details that others could replicate the analyses.

For instance, if you developed a questionnaire as part of this study and it is not under a copyright more restrictive than CC-BY, please include a copy, in both the original language and English, as Supporting Information.

5. Please upload a copy of Figure 5, to which you refer in your text on page 9. If the figure is no longer to be included as part of the submission please remove all reference to it within the text.

Additional Editor Comments:

This manuscript is important and interesting. Please revise your manuscript according to the reviewer concerns.

Reviewers' comments:

Reviewer's Responses to Questions

**Comments to the Author**

1. Is the manuscript technically sound, and do the data support the conclusions?

Reviewer #1: Yes

2. Has the statistical analysis been performed appropriately and rigorously? 

Reviewer #1: Yes

3. Have the authors made all data underlying the findings in their manuscript fully available?

Reviewer #1: Yes

4. Is the manuscript presented in an intelligible fashion and written in standard English?

Reviewer #1: Yes

5. Review Comments to the Author

Reviewer #1: This is an important piece of research which throws light on how the situations arising out of COVID 19 affects individual well-being in a comprehensive way in South Australia. The research shows how the pandemic has cast multi-dimensional impacts on individuals’ well-being along with significant drivers of well-being. The paper suggests that while the majority of people in South Australia survive the ill-effect of the pandemic, about a third of the population is found to have been found to be “multi-dimensionally” affected with more than 3 dimensions of out of six being adversely affected simultaneously. This stresses upon the fact that the current pandemic is not only a health-crisis, rather more a social crisis. The paper goes on to argue that impacts of the pandemic have distinct gradients in terms of economic status, age and attitude towards life. The survey results reveal that the impacts have been disproportionately extreme in case of the young, poor and those with a weak attitude to life. Since, these are crucial survey findings, it will be better if the results of “formal statistical tests” so that these observed tendencies can be generalised.

The paper adopts a multi-dimensional framework in analysis with Alkire-Foster method as the guiding basis. The analysis is based double-cut-offs applying two-third rule and identification of the “multi-dimensionally” experiencing ill-being has been done in case of individuals who suffer in at-least 4 out of 6 dimensions together. The paper provides that 167 out of 579 are such individuals (i.e. 29 percent). This is the head-count. It will also be interesting to examine the “intensity” of ill-being – in the line of Alkire-Foster method – which will indicate the average number of dimensions that are being suffered by these 167 individuals. Table 4, similarly, gives the ‘censored percentages’. The paper may compare this by taking note of the “uncensored percentages” to see how different dimensions play out in the population as a whole.

I was just wondering whether individuals can be identified who have suffered from COVID 19 in the database. If we can have data on whether the individuals have suffered from COVID 19, or whether any of their family/relatives suffered from COVID 19 the results can further be disaggregated to see how they vary across these categories.

In summary, the paper is a valuable contribution and is recommended for publication with the above minor observations.

6. PLOS authors have the option to publish the peer review history of their article (what does this mean?). If published, this will include your full peer review and any attached files.

Reviewer #1: **Yes: **Joydeep Baruah

---

## [Author Response · Author response to Decision Letter 0]

16 May 2021

PONE-D-21-01771

COVID-19, individual wellbeing and multi-dimensional poverty in the state of South Australia 

PLOS ONE

Response to Reviewers

07/05/2021

Dear Sir/Madam,

As requested in your email of 28 April 2021, below please find a response to each of the points raised by the academic editor and reviewer of the manuscript indicated above. 

Academic Editor Comments

Authors’ Response: Manuscript and associated files have been updated accordingly.

Authors’ Response: The reference list is complete, correct, and has been updated in the final Manuscript to reflect the required referencing and citation format. Please note these formatting changes are not reflected in the Revised Manuscript with Track Changes so as to reduce unnecessary confusion with the content changes in response to the Reviewers comments. 

3. Please include additional information regarding the survey or questionnaire used in the study and ensure that you have provided sufficient details that others could replicate the analyses.

Authors’ Response: The questionnaire associated with this study has been provided as S1 File.

Authors’ Response: We have been advised by Flinders Research Ethics and Compliance that as the participants did not consent to sharing the data with other researchers/organisations, we are unable to provide it here, even in a de-identified form. Flinders Research Ethics and Compliance can be contacted at human.researchethics@flinders.edu.au.

5. Please upload a copy of Figure 5, to which you refer in your text on page 9. If the figure is no longer to be included as part of the submission please remove all reference to it within the text.

Authors’ Response: There is no Figure 5 associated with this manuscript. All references to such Figure have been modified or removed. 

Additional Editor Comments:

This manuscript is important and interesting. Please revise your manuscript according to the reviewer concerns.

Reviewers' comments:

Reviewer's Responses to Questions

Comments to the Author

1. Is the manuscript technically sound, and do the data support the conclusions?

Reviewer #1: Yes

Authors’ Response: None required.

2. Has the statistical analysis been performed appropriately and rigorously? 

Reviewer #1: Yes

Authors’ Response: None required.

3. Have the authors made all data underlying the findings in their manuscript fully available?

Reviewer #1: Yes

Authors’ Comment: We have been advised by Flinders Research Ethics and Compliance that as the participants did not consent to sharing the data with other researchers/organisations, we are unable to provide it here, even in a de-identified form. Flinders Research Ethics and Compliance can be contacted at human.researchethics@flinders.edu.au.

4. Is the manuscript presented in an intelligible fashion and written in standard English?

Reviewer #1: Yes

Authors’ Response: None required.

5. Review Comments to the Author

Reviewer #1: This is an important piece of research which throws light on how the situations arising out of COVID 19 affects individual well-being in a comprehensive way in South Australia. The research shows how the pandemic has cast multi-dimensional impacts on individuals’ well-being along with significant drivers of well-being. The paper suggests that while the majority of people in South Australia survive the ill-effect of the pandemic, about a third of the population is found to have been found to be “multi-dimensionally” affected with more than 3 dimensions of out of six being adversely affected simultaneously. This stresses upon the fact that the current pandemic is not only a health-crisis, rather more a social crisis. The paper goes on to argue that impacts of the pandemic have distinct gradients in terms of economic status, age and attitude towards life. The survey results reveal that the impacts have been disproportionately extreme in case of the young, poor and those with a weak attitude to life. Since, these are crucial survey findings, it will be better if the results of “formal statistical tests” so that these observed tendencies can be generalised.

The paper adopts a multi-dimensional framework in analysis with Alkire-Foster method as the guiding basis. The analysis is based double-cut-offs applying two-third rule and identification of the “multi-dimensionally” experiencing ill-being has been done in case of individuals who suffer in at-least 4 out of 6 dimensions together. The paper provides that 167 out of 579 are such individuals (i.e. 29 percent). This is the head-count. It will also be interesting to examine the “intensity” of ill-being – in the line of Alkire-Foster method – which will indicate the average number of dimensions that are being suffered by these 167 individuals. Table 4, similarly, gives the ‘censored percentages’. The paper may compare this by taking note of the “uncensored percentages” to see how different dimensions play out in the population as a whole.

I was just wondering whether individuals can be identified who have suffered from COVID 19 in the database. If we can have data on whether the individuals have suffered from COVID 19, or whether any of their family/relatives suffered from COVID 19 the results can further be disaggregated to see how they vary across these categories.

In summary, the paper is a valuable contribution and is recommended for publication with the above minor observations.

Authors’ Response:

The results of the ‘formal statistical tests’, including income, attitude toward life, and age, can be found in Table 3: Socio-economic determinants of wellbeing – results of binomial logistic regression. 

In response to the reviewer’s comment regarding ‘intensity’, the manuscript has now been modified to include additional discussion of this aspect of the research (pg. 15). Similarly, Table 4 now includes a column showing the ‘uncensored percentages’ for comparison. 

With regard to identifying individuals who have suffered from COVID-19, this is not possible given the design of the survey, which did not ask participants to disclose this information. 

6. PLOS authors have the option to publish the peer review history of their article (what does this mean?). If published, this will include your full peer review and any attached files.

Do you want your identity to be public for this peer review? For information about this choice, including consent withdrawal, please see our Privacy Policy.

Reviewer #1: Yes: Joydeep Baruah

Authors’ Response: None required. ________________________________________

Should you need additional information or further clarification please do not hesitate to contact me.

Thank you for your time and interest in publishing this manuscript.

---

## [Decision Letter · Decision Letter 1]

25 May 2021

COVID-19, individual wellbeing and multi-dimensional poverty in the state of South Australia

PONE-D-21-01771R1

Dear Dr. Saikia,

We’re pleased to inform you that your manuscript has been judged scientifically suitable for publication and will be formally accepted for publication once it meets all outstanding technical requirements.

Kind regards,

Rashidul Alam Mahumud, MPH, MSc, PhD

Academic Editor

PLOS ONE

Additional Editor Comments (optional):

Reviewers' comments:

Reviewer's Responses to Questions

**Comments to the Author**

1. If the authors have adequately addressed your comments raised in a previous round of review and you feel that this manuscript is now acceptable for publication, you may indicate that here to bypass the “Comments to the Author” section, enter your conflict of interest statement in the “Confidential to Editor” section, and submit your "Accept" recommendation.

Reviewer #1: All comments have been addressed

2. Is the manuscript technically sound, and do the data support the conclusions?

Reviewer #1: Yes

3. Has the statistical analysis been performed appropriately and rigorously? 

Reviewer #1: Yes

4. Have the authors made all data underlying the findings in their manuscript fully available?

Reviewer #1: Yes

5. Is the manuscript presented in an intelligible fashion and written in standard English?

Reviewer #1: Yes

6. Review Comments to the Author

Reviewer #1: The revised paper has accommodated the comments made on the original submission and I am satisfied with the authors responses. The revised paper now may kindly be accepted for publication.

7. PLOS authors have the option to publish the peer review history of their article (what does this mean?). If published, this will include your full peer review and any attached files.

Reviewer #1: **Yes: **Joydeep Baruah

---

## [Editor Report · Acceptance letter]

3 Jun 2021

PONE-D-21-01771R1 

COVID-19, individual wellbeing and multi-dimensional poverty in the state of South Australia 

Dear Dr. Saikia:

I'm pleased to inform you that your manuscript has been deemed suitable for publication in PLOS ONE. Congratulations! Your manuscript is now with our production department. 

Kind regards, 

on behalf of

Dr. Rashidul Alam Mahumud 

Academic Editor

PLOS ONE